# Bedload Transport Monitoring in Alpine Rivers: Variability in Swiss Plate Geophone Response

**DOI:** 10.3390/s20154089

**Published:** 2020-07-22

**Authors:** Gilles Antoniazza, Tobias Nicollier, Carlos R. Wyss, Stefan Boss, Dieter Rickenmann

**Affiliations:** 1Swiss Federal Institute for Forest, Snow and Landscape Research (WSL), Mountain Hydrology and Mass Movements, Zürcherstrasse 111, CH-8903 Birmensdorf, Switzerland; tobias.nicollier@wsl.ch (T.N.); carlos.wyss@wsl.ch (C.R.W.); stefan.boss@wsl.ch (S.B.); dieter.rickenmann@wsl.ch (D.R.); 2Institute of Earth Surface Dynamics (IDYST), University of Lausanne (Switzerland), Géopolis, CH-1015 Lausanne, Switzerland

**Keywords:** bedload transport, acoustic sensors, Alpine stream, Swiss plate geophone system (SPG), impact experiment, sensor sensitivity, sensor calibration

## Abstract

Acoustic sensors are increasingly used to measure bedload transport in Alpine streams, notably the Swiss plate geophone (SPG) system. An impact experiment using artificial weights is developed in this paper to assess the variability in individual plate response and to evaluate the extent to which calibration coefficients can be transferred from calibrated plates to non-calibrated plates at a given measuring site and/or to other measuring sites. Results of the experiment over 43 plates at four measuring sites have notably shown (a) that the maximum amplitude (V) recorded by individual plates tends to evolve as a power law function of the impact energy (J), with an exponent slightly larger than 1, for all the plates at all measuring sites; (b) that there is a substantial propagation of energy across plates that should be taken into account for a better understanding of the signal response; (c) that the response of individual plates is in most cases consistent, which suggests that calibration coefficients are comparable within and in between measuring sites, but site-specific and plate-specific variabilities in signal response have to be considered for a detailed comparison.

## 1. Introduction

Bedload transport in rivers represents the motion of coarse particles of the bed by rolling, sliding and saltation when the shear force of the water flow exceeds the resistance force of the bed matrix [1,2,3]. As such, bedload is both a consequence of fluvial erosion and a driver for fluvial deposition [3,4,5]. Bedload has therefore a critical impact on flooding risk [6,7,8], can induce bank erosion and in turn impact infrastructures [9,10]. It can also reduce the efficiency of hydropower facilities [11,12,13,14] and impact fluvial ecosystems both negatively and positively [15,16]. Bedload transport therefore matters for many ecosystems and human activities in the vicinity of rivers, notably in Alpine environments where the rate of climate warming is accelerated and leads to rapid glacier recession and/or an increase in the frequency of extreme hydrological events, which both tend to enhance bedload activity [17,18].

At the same time, bedload transport remains one of the most elusive phenomena to measure and predict. This is notably due to the complex interactions between the flowing water, the heterogeneous nature of riverbed sediments, the unsteady hydraulics (themselves being a product of mobile channel boundaries), and the sediment transport [4,19,20,21]. These elements particularly matter in Alpine environments where high and spatially-variable gradients in energy are observed, associated to the complex effects of coarse roughness elements on the stream bed, the turbulent and unsteady hydraulics that result, and the frequent reworking of stream bed by erosion and deposition [22,23,24,25,26].

Early measurements of bedload have used direct sampling with baskets or pressure-difference samplers [27,28]. Discrete in time and at a point in space, they however failed to quantify the spatio-temporal variability in bedload transport [20,21]. Meanwhile, measurements of the filling rate of traps dug across streams have provided cross-section integrated estimates of bedload transport rate [29,30,31]. However, such traps are often rapidly filled, they affect the hydraulic conditions and provide no detail on the temporal (within events) and spatial (across the section) variations of bedload transport [20,32]. Progress with passive acoustic sensors over the past 20 years opened new perspectives for the continuous monitoring of bedload transport [33,34,35,36]. Bedload produces acoustic (in the water column) and elastic (in media such as the riverbed) waves that are measurable with acoustic sensors. The magnitude of these waves tends to be proportional to bedload transport rates. Therefore, acoustic sensors deployed in a river and measuring the sounds (e.g., hydrophones, microphones) or the vibrations of structures impacted by particles in transport (e.g., geophones, accelerometers, piezoelectric sensors) are able to monitor continuously the bedload transport intensity, without modifying the surrounding hydraulic conditions. When a network of acoustic sensors is deployed (along and/or across a channel), both spatial and temporal variations of bedload transport intensity can be monitored. This approach has been successfully used in many environments, including steep and highly-energetic Alpine streams [37,38,39,40,41], for both theoretical [38,39,40,42,43] and applied [37,44,45,46] purposes. By providing an indirect signal of bedload transport intensity, acoustic sensors however require calibration and validation to transform the signal recorded by the sensors into an actual mass of bedload in transport [36,38,40].

Among acoustic sensors, the Swiss plate geophone (SPG) system developed by the Swiss federal Institute WSL is one of the acoustic measuring systems with the most widespread experience to monitor bedload transport in Alpine streams [36,37,38]. Over the past 20 years, more than 20 measuring stations have been equipped with such a measuring system in Switzerland, Austria, Italy, Israel, and in the United States [36]. The SPG system is made of a series of steel plates embodied in a concrete structure flush with the riverbed to not affect surrounding hydraulic conditions. Particles rolling, sliding, and saltating over the bed impact the steel plates, and a geophone sensor mounted from below each plate continuously records the induced vibration. SPGs are typically mounted in a line across the stream to cover with sensors the entire channel width [44,45]. However, like any other acoustic measuring system, the SPG requires calibration and validation to transform the recorded signal into an actual mass of bedload in transport. This is typically performed by collecting direct bedload samples that can be related to the signal measured by the sensors [37,38,44]. Such calibration and validation samples are typically difficult to obtain and require the design of robust sediment baskets that can be deployed in Alpine streams during high flows. This has been achieved so far through two main approaches. First, by using mobile sediment baskets mounted on a rail, which is built along a concrete sill and allows to move the sediment basket across the section to collect bedload particles that have impacted the plates [37,38]. Second, by using sediment baskets or net samplers mounted on a crane and operated from a truck on the riverbank, which can collect bedload samples downstream of the impacted plates [37,44,47,48,49].

However, the collection of sufficient bedload samples for each plate at a given measuring site is highly demanding in terms of field effort, infrastructure, cost, and time. Therefore, it is important to assess the variability in signal response between different plates and different sites. In doing so, the extent to which calibration coefficients can be transferred from calibrated plates to uncalibrated ones can be evaluated. The general objective of this paper is to assess the variability in signal response of SPGs to impacts of similar magnitude. For this purpose, we performed an experiment using synthetic weights that are employed to artificially generate an impact on the plate, which is recorded by the geophone sensor. The impact experiment was conducted on each plate at four measuring stations in the Swiss Alps equipped with the SPG system. In particular, we were interested to examine (i) how variable the signal response is between individual plates within and in between the measuring stations, and (ii) how much energy propagates from an impacted plate towards non-impacted plates. These two elements matter because, ultimately, they both influence the calibration relation obtained for a given site and plate, and the potential transferability of calibration relations to other (uncalibrated) plates.

## 2. Materials and Methods

The aim of the experiment developed in this paper is to study the response variability of Swiss plate geophones. To do so, an impact test using artificial weights was performed on 43 plates spread over four sites. First, we describe the design of both the weights and the impact experiment. Then, we report on the experiment using a laboratory SPG. Finally, we describe the application of the impact experiment at the four field sites.

### 2.1. Material

The impact experiment developed in this paper was performed as follows. Impacts were produced by using a generic steel pedologist hammer (Figure 1a), whose usual purpose is to pack down soils by letting an artificial weight fall down along a stake on the hammer base, which lies on the soil surface. The weight (Figure 1b) hits the hammer base and energy is transmitted through it to the ground (Figure 1c). During the experiment, the apparatus was set at the exact center (uncertainty ± 0.01 m) of each tested plate (Figure 1d), and a minimum support pressure was given to the stake to not add any unnecessary random tension to the plate beside the mass of the device itself (2.272 kg + mass of the released weight (kg); Table 1). 

The SPGs are made from stainless steel and have standard dimensions of 492 mm width, 358 mm length (in flow direction), and 15 mm thickness [36,38,40], resulting in a mass of 21 kg. The weights used in the impact experiment are composed of PVC cylinders with the same diameter as the hammer base (0.05 m). Central holes of 0.02 m diameter were drilled trough the PVC weights to make them fall down along the metal stake. The diameter of the central hole is slightly larger than the diameter of the stake (0.015 m) to reduce the risk of friction when falling. Then, each PVC cylinder was cut at the height matching the wanted mass (Table 1). As most of the tested plates were underwater during the experiment (up to 0.1 m), it was necessary to add a PVC cylinder of 0.1 m height above the base (so-called ‘heightening cylinder’) and seal them together to make sure the weight would hit the base above water level. Striking the plates from above the water surface represents a limit as it is not likely to mimic how sediment actually impacts the plates in natural flow conditions. Yet, the field tests were performed at varying low-flow conditions (from no-water to flowing water up to ≈0.1 m), and thus there is a partial similarity to natural transport conditions.

The setting of the impact experiment developed in this paper also includes some sources of uncertainty. Both the surrounding air and the potential friction of the weight along the stake on the way down can affect the delivered energy. In addition, the propagation of the energy from the weight to the plate through the base can increase the variability in the transmitted energy for an impact assumed to be of same magnitude. Therefore, an initial laboratory experiment was performed to better constrain the experiment and make sure it was reproducible. In addition, video recordings of the falling weights were taken using a high-speed camera in order to quantify the velocity of the weight at the impact and the associated kinetic energy delivered for each weight-height drop category.

### 2.2. Laboratory Experiment

#### 2.2.1. Reproducibility Test

The initial laboratory experiment was performed on a single plate installed at the Laboratory of Hydraulics, Hydrology and Glaciology (VAW) of the ETH Zürich [39,40] in a similar configuration as the SPGs installed in the field.

In a first phase of the experiment, a reproducibility test was conducted by impacting the laboratory plate with a medium weight mass—0.064 kg—dropped from a 0.1 m falling height, and repeating the same procedure 25 times. The maximum positive amplitude of the signal A_max_ (V) as defined by Wyss et al. [41] was recorded for each impact. The mean, standard deviation, 5th and 95th percentile of the experiment after k = 5, 10, 15, 20, and 25 impacts were calculated out of 5 million random combinations (k tests out of the total 25 tests). Results (Figure 2a) show that the standard deviation of the impacts stabilizes when the amount of impacts reaches ≈10. Based on this knowledge, we decided to repeat a same impact 10 times per category of weight-height to take into account the variability in energy transmission from the impacting weight to the plate through the hammer base when conducting the field experiment. An example of a given category of impact (in terms of weight and height) repeated 10 times, such as performed in the field, is presented in Figure 1b (here for a 0.064 kg weight dropped from a 0.1 m height).

#### 2.2.2. Experiment Design

Although the shape of the weights is unlikely to closely mimic the impact of natural bedload particles, we wanted the impacts of the experiment to cover a range of voltage that is close to the one recorded by the SPG for natural bedload events. Earlier works showed they were typically ranging between 0.02 and 10 V in Alpine rivers equipped with the SPG system [38,40]. To fulfill this condition, initial tests in the laboratory were carried out with nine different categories of weight-height according to Table 2. Both the maximal amplitude of the impacts $\bar{A_{\max}}$ (V) (mean of 10 impacts) and their standard deviation $S_{x}\;A_{\max}$ (V) (standard deviation of 10 impacts) tend to increase with the energy released by a given combination of weight-height (Table 2). The variability between the impacts of a given drop category tends however to remain relatively low, between 6% and 26% of the mean maximum amplitude $\bar{A_{\max}}$. Given the uncertainty related to the energy transmitted by the impacting weight to the plate through the hammer base, the variability in the recorded A_max_ between impacts of same theoretical magnitude appears to be acceptable.

#### 2.2.3. Kinetic Energy of the Impacts

As the likeliness of having friction along the stake is greater for longer fall distance (i.e., falling height of 0.6 m as compared to 0.1 m), we investigated the actual kinetic energy produced on average for each category of weight-height drops. To do so, we performed videos in the laboratory using a high-resolution AOS X-PRI high-speed camera (AOS Technologies, Baden-Daettwil, Switzerland) and filmed the speed at which the different categories of weight-height impacted the base of the pedologist hammer. We considered the last 5 ms before the impact, and repeated each fall three times. The kinetic energy was derived from the falling speed before the impact, as: (1)$$E_{k}=0.5*m*v^{2}$$
where *E_k_* is the kinetic energy of the impact (J), m is the mass of the weight (kg), and v (m/s) is the speed of the weight during the last 5 ms before the impact. Table 3 summarizes the falling speed and kinetic energy for each category of weight-height (mean of three drops). As expected, the fall velocity and the kinetic energy of the weights increase together with the weight mass and the height of the drop. 

The calculation of the kinetic energy allows to investigate the relationship with the A_max_ recorded for each category of impact, which seems to be best expressed by a power law function, with an exponent slightly above 1 (Figure 3). In the laboratory, the experiment covers an energy field between 0.006 and 1 J, and A_max_ ranges between 0.04 and 10 V, which fits well the targeted range of amplitudes that is observed for natural bedload transport events in Alpine rivers equipped with the SPG system [38,40].

### 2.3. Field Experiment

After testing the applicability of the method in the laboratory, each plate at four Swiss field measuring stations was impacted 10 times for each of the nine categories of weight-height drop presented in Table 3, which means that each plate was impacted 90 times. Each of those measuring stations was constructed in a similar way: a concrete sill across the channel width was built flush with the riverbed, and a steel frame is embedded into the sill. The plates are then mounted within the steel frame, and 20DX geophone sensors from Geospace technologies (Houston, Texas, USA) are mounted from below the plate in a PC801 LPC Landcase set in an aluminum box [45]. During sediment transport events, bedload particles sliding, rolling, or saltating over a given steel plate produces an impact shock that induces a change of the electrical voltage measured by the geophone, which is assumed to be proportional to the amount of bedload in transport [38,40]. Each plate is acoustically isolated from each other by elastometer elements, and the geophone sensors are connected to an industrial PC that records the signal at a frequency of 10 kHz. The amount of plates per measuring station is set so that the greatest portion of the cross-section is covered by sensors. 

In the frame of this paper, we aim to quantify the variability in plate response on the one hand, but also the extent to which there may be some energy that propagates from impacted plates towards non-impacted plates. Indeed, earlier analyses of the SPG system [37,38,40] have suggested that there may be some energy that propagates across plates despite the isolating elastomer elements. This needs to be quantified properly to know how it influences the relationship between the signal recorded by the plate and the actual bedload mass in transport on the one hand, and the extent to which calibration coefficients could be used more broadly within bedload measuring stations equipped with the SPG system on the other hand. At each measuring station, SPGs are grouped together into steel frame segments holding 5–6 plates with sensors. The specific setting of each measuring station in terms of number of plates and segments is presented in Table 4 and Figure 4.

The operation of the measuring station at the Avançon de Nant (Western Swiss Alps) started in 2014 and results from a collaboration between the University of Lausanne (UNIL), the Swiss Federal Institute for Forest, Snow and Landscape Research (WSL) and the ETH-Zürich. The station is equipped with 10 SPGs (Figure 4a), installed in two different segments. When the experiment was performed in October 2015, the flow discharge was very low and there was only ≈0.02 m of water flowing over the concrete weir. Therefore, no additional random tension was added to the pedologist hammer by the need of holding it, and the experiment could be conducted with a minimal disturbance of the measuring system, close to laboratory conditions.

The measuring station at the Albula Alpine river (Eastern Swiss Alps) was built in 2015. Over the 15 m width of the channel at the location of the station (Figure 4b), 30 impact plates were installed, below which 15 geophones and 15 accelerometers were alternatively set up. The impact plates at the Albula are connected into five segments, gathering six sensors each (three geophones and three accelerometers). In the frame of this project, the experiment was only conducted on the SPGs to make the results comparable to the datasets from other stations where only SPGs are mounted. The experiment was conducted on every SPG but n°1 (omission). When we performed the experiment in November 2015, the discharge was fairly low, but there was still about ≈0.1 m of water depth over the sensors flowing at ≈0.3–0.4 m/s, which forced us to slightly support the pedologist hammer by holding the stake during the experiment. However little the support was, it may have added some additional tension to the plates.

The measuring station at the Naviscence (Southern Swiss Alps) was set in operation in 2011. Twelve SPG sensors are mounted flush with the riverbed within the concrete sill (Figure 4c). On the day the experiment was conducted in October 2015, conditions were similar to the ones we encountered at the Albula, which means that some support was required for the pedologist hammer during the experiment, which may have also added some additional tension to some of the plates. SPGs at the Naviscence are mounted into two segments gathering six sensors each. 

The measuring station at the Riedbach (Southern Swiss Alps) was set in operation in 2009 and is equipped with seven SPG sensors, which are slightly inclined (i.e., 5–10°) towards the slope (Figure 4d). This means that even if there was no flowing water over the sensors at the day of the experiment in October 2015, the inclination of the plates may have influenced the way the weights were falling and impacting the plates. This may also have consequences on the comparability of the Riedbach measuring station with the results coming from the other measuring stations where the SPGs are mounted over a flat sill. The seven SPG sensors at the Riedbach are connected to a single segment.

### 2.4. Metric Calculation

Datasets of the impact experiment were used to calculate three main elements. First, the maximum positive amplitude A_max_ (V) that each impact produces on a given plate [37,38,40,50]. Second, the amount of energy from a given impact that propagates in space to neighboring non-impacted plates, also measured as the maximum positive amplitude A_max_. Third, the number of impulses N_imp_ (-), which is the number of times the raw signal exceeds a predefined threshold. This threshold is set at 0.1 V at the four SPG measuring stations, because it was estimated to be greater by at least one order of magnitude as compared to the noise level for conditions without bedload transport [39,40]. Data are then used to quantify (a) the response of each individual plate at the four measuring stations and (b) the main characteristics of the energy that propagates from impacted plates towards non-impacted plates. We then draw from (a) and (b) to generalize findings about the response variability of SPGs, and its implications regarding the necessary calibration effort.

## 3. Results

### 3.1. Direct Maximum Amplitude

The relationship between *E_k_* and the recorded A_max_ is plotted in Figure 5 for each individual plate at (a) the Avançon de Nant, (b) the Albula, (c) the Naviscence, and (d) the Riedbach. The kinetic energy values were taken from the laboratory experiment, and the A_max_ values correspond to the energy recorded directly on a given impacted plate. In Figure 5, each dot represents the mean of a given drop category (in terms of impact energy) repeated 10 times. The intercept, slope, and R^2^ of the regressions for all the plates from all tests at one measuring station taken together are given in Table 5. The regression coefficients for every individual plate are reported in the Appendix A. Such as observed in the laboratory experiment (Figure 3), the energy resulting from the different categories of drops resulted in A_max_ ranging from ≈0.01 to ≈10 V. A_max_ tends to increase as a power law function of the kinetic energy delivered by the impact for all the plates at all four measuring stations, the exponent of the power law relationship being on average slightly larger than 1, such as observed for the laboratory experiment. However, some signal variability is visible between plates of a given measuring station. The variability is more pronounced at the Albula and Naviscence measuring stations (factor 3-5 of difference), while it is smaller at the Avançon de Nant and Riedbach measuring stations (factor 1-2 of difference). 

### 3.2. Energy Propagation Across Plates

We also quantified the extent to which some energy propagates in space from impacted plates towards non-impacted plates. 

As the Riedbach measuring station is setup with an older electronic measuring system that does not allow for recording the signal over all the plates simultaneously, this analysis could only be performed for the other three measuring sites.

Results from Figure 6 illustrate that the energy tends to propagate to some extent from impacted plates towards non-impacted plates within a given segment, but poorly across different segments. This is notably true at the Avançon de Nant and Albula measuring stations (Figure 4a,b). For these two stations, the propagation of the signal was therefore quantified for the plates placed within the same segment. Depending on the position of an impacted plate within a segment (in the middle or at an edge), the A_max_ on the neighboring plates at 1–4 plates of distance was measured from a single plate, or as a mean between two plates. For instance, when impacting SPG 4 at the Avançon de Nant measuring station, there are two plates located at one plate of distance (SPGs 3 and 5), but only one located at two plates of distance (SPG 2) when considering the propagation of the energy within the same segment. In the first case, the A_max_ recorded at one plate of distance is calculated as the mean between the A_max_ of SPG 3 and SPG 5, while at two plates of distance it is computed as the A_max_ from SPG 2 only. At the Naviscence, the energy seems to also propagate across the different segments (Figure 6c), and only at this site the energy propagation was therefore also assessed across the two neighboring segments. For instance, the amount of propagated energy when impacting SPG 6 at the Naviscence was taken as the mean of SPG 5 and SPG 7, even if SPG 7 is installed into another segment. 

In Figure 7, Figure 8 and Figure 9, the energy propagation towards neighboring plates is expressed in terms of A_max_ and as a relative reduction of A_max_. As there are accelerometers mounted below each second plate at the Albula measuring station, the energy propagation from impacted plates towards non-impacted plates in this station could only be computed at two and four plates of distance, respectively, when considering each segment separately (Figure 6 and Figure 8). As can be expected, the amount of energy that propagates from impacted plates toward non-impacted plates at all three sites increases with the magnitude of the impact. The energy propagation across plates can be expressed by a power law function, with an exponent slightly lower than 1, which decreases progressively as the energy propagates further away from the impacted plate. Exponents are steadily slightly greater at the Avançon de Nant (Figure 7) as compared to the Albula (Figure 8) at both two and four plates of distance, and the rate of attenuation is much faster at the Naviscence as compared to the other two sites (Figure 9) from two plates of distance on. At the Albula and Naviscence measuring sites, the propagated energy across plates does not increase significantly for low magnitude A_max_. It only does so when a threshold of 0.02 V is exceeded at the Albula, and thresholds of 0.08, 0.5, and 1 V are exceeded at the Naviscence at 1–4 plates of distance, respectively. In relative terms, the proportion of the energy that propagates from impacted plates towards non-impacted plates is however greater for small magnitude impacts at all three sites. In summary, in terms of A_max_, a similar level of propagation is observed at the Avançon de Nant and Albula sites, whereas the propagated values are clearly smaller at the Navisence site (Figure 10).

Considering the threshold of impulse counting of 0.1 V, which is the reference at many measuring stations equipped with the SPG system [37,38,40], impacts greater than 0.6 V are required to exceed the 0.1 V threshold at one plate of distance at the Avançon de Nant measuring station. Impacts greater than 1 and 2 V are required to exceed it at two, three, and four plates of distance, respectively (Figure 7, Table 6). At the Albula, impacts greater than 1 V are required to exceed the 0.1 V threshold at two plates of distance, and greater than 2 V at four plates of distance (Figure 8, Table 6). At the Naviscence, only impacts greater than 1 V produce a signal that exceeds the 0.1 V threshold at one plate of distance. At two, three, and four plates of distance, even the strongest impacts do not produce a signal that exceeds 0.1 V, at least in the investigated domain (impacts up to 10 V) (Figure 9, Table 6).

Once the 0.1 V threshold is exceeded, the proportion of the impulse count N_imp_ that is accounted for on the first, second, third, and fourth neighboring plates remains however steadily higher than the proportion of A_max_. At the Avançon de Nant, it represents on average 55%, 49%, 46%, and 55% at one, two, three, and four plates of distance, respectively (Figure 10b). At the Albula, it represents on average 39% and 32% of the impulses counted on the impacted SPG at two and four plates of distance (Figure 10b). At the Naviscence at one plate of distance, 37% of the impulses remain on average accounted for (Figure 10b). The relationship between the A_max_ and the impulse count N_imp_ (>0.1 V) at the Avançon de Nant, Albula, and Naviscence measuring stations are presented in a Appendix A.

As observed in Figure 7, Figure 8 and Figure 9, there is an attenuation of the propagated energy across plates with distance from the impacted plate when considering the A_max_. In Figure 11, the regression equations between the impact A_max_ and the propagated A_max_ across plates were fitted for all the impacts on every plate taken together for a given impact distance (i.e., one, two, three, or four plates of distance) and at a given site. At the Avançon de Nant, there is a significant attenuation of energy propagation between one and two plates of distance (factor 2), then smoother between two and three plates of distance (factor 0.5), then there is no more attenuation between three and four plates of distance. At the Albula, the attenuation between two and four plates of distance is moderate (factor 1/3–2/3). At the Naviscence, the attenuation is substantial between one and two plates of distance, about a factor 2.5 for impacts of 0.5 V to a factor 8.5 for impacts close to 10 V. Then the attenuation decays between two, three, and four plates of distance (to a factor 2–3), but the remaining propagated energy is very low.

### 3.3. Sensor Sensitivity

Comparing the regression lines of the relationship between the impact A_max_ and the propagated A_max_ across plates (here plotted for one and two plates of distance), also allows to investigate whether the different plates have a similar sensitivity, or whether there is a significant response variability (a) between different plates of a given measuring station and (b) between plates at different measuring sites (Figure 12), to energy propagation across plates. To estimate the individual sensor sensitivity in Figure 12, the propagation of energy for one plate was calculated slightly differently as compared to Section 3.2. Here, instead of taking the mean value of the propagated energy of a given impact (measured as a A_max_) on the neighboring plate(s), we took the mean of the propagated energy (measured as a A_max_) from the neighboring plate(s) onto a given plate. This explains the small differences that can be observed in power law coefficients between Section 3.2. (e.g., Figure 7, Figure 8, Figure 9, Figure 10 and Figure 11) and Section 3.3. (e.g., Table 7). The power law coefficients of all impacts on every plate at a given site are given in Table 7. The power law coefficients of every individual plate at the three measuring sites are reported in the Appendix A.

In terms of (a), it appears that the 10 plates at the Avançon de Nant measuring station have a comparable response at both one and two plates of distance (Figure 12a), with exponents of the power law relationship slightly below 1 (Table 7). At the Albula, the variability between the different plates is slightly larger, while the exponent of the power law function is of a comparable magnitude with the Avançon de Nant (Table 7). At the Naviscence, the variability between the different plates is the largest, both considering the coefficient (factor 2–3) and the exponent of the power law functions (between 0.786 and 1.05 at one plate of distance, and between 0.661 and 0.946 at two plates of distance, Appendix A).

In terms of (b), the plates at the Avançon de Nant seem to be overall more sensitive to energy propagation across plates, than at the Naviscence (about a factor 2) at one plate of distance. The between-plates variability at a given measuring station is smaller than the between-site variability for the Avançon de Nant and Naviscence measuring stations at one plate of distance. At two plates of distance, the Avançon de Nant and the Albula have a comparable response, and the between-plates variability cannot be clearly distinguished from the between-site variability. The pattern at the Naviscence is clearly different due to the greater rate of signal attenuation, and the between-plates variability is lower than the between-site variability.

## 4. Discussion

### 4.1. Comparability in SPG Response and Energy Propagation Across Plates

The impact experiment presented in this paper allowed to determine the response of 43 SPGs at four measuring sites to impacts of comparable magnitude, and to quantify the amount of energy that propagates in space from impacted plates towards non-impacted plates. Results from the field were similar to the initial laboratory experiment. The relation between A_max_ and *E_k_* is best expressed by a power law with an exponent on average slightly larger than 1 for all the sensors at all measuring sites (Figure 3 and Figure 5, Table 5, Appendix A). This result is encouraging because it indicates that all sensors respond in a similar way to impacts of increasing magnitude.

Regarding energy propagation across plates, the proportion of A_max_ that is recorded on neighboring non-impacted plates is substantial, and should notably be accounted for at one and two plates of distance, whereas the propagated energy attenuates much more at larger distance away from the impact location. When considering the impulse count N_imp_ (>0.1 V), the propagated impulses from impacted to non-impacted plates is even greater and does not attenuate with distance to the same extent as A_max_. Therefore, the spatial propagation of energy from impacted plates towards non-impacted plates should be considered for an entire segment when using the impulse count as an indicator of bedload transport intensity. 

The energy propagation across plates can be reasonably well expressed by a power law function between A_max_ at the impacted plate and the propagated A_max_ for all the plates and at all three investigated sites. Exponents of the power law relationship at one plate of distance were slightly smaller than 1, and decreased with distance from the impacted plate due to attenuation effects. Considering the sensitivity of individual SPGs to energy propagation across plates, every non-impacted plate at the Avançon de Nant and Albula measuring sites seem to record a comparable amount of energy for a given impact A_max_ on a neighboring plate, which suggest that the plates at these two sites have a comparable sensitivity to energy propagation across plates (Figure 11, Appendix A). At the Naviscence, the rate of attenuation is greater but the slope of the relationship remains comparable in between the three sites (Figure 11, Appendix A).

### 4.2. Variability in SPG Response and Energy Propagation across Plates

Although the SPG response and the energy propagation across plates showed general similarities between the investigated sites, a non-negligible variability in the signal response was observed among individual plates at a given site, and in between sites. This was notable (a) from a comparison of the power law coefficients of the relations between A_max_ and *E_k_* between individual plates and (b) in the propagation of energy across plates. As compared to the Avançon de Nant and Albula plates, the SPGs at the Naviscence notably presented a lower level of sensitivity to energy propagation across plates, higher A_max_ needed to exceed the impulse count threshold of 0.1 V on non-impacted plates, so as a greater rate of attenuation with distance from the impact. Meanwhile, the propagation of energy across SPG segments was shown (c) to be negligible at the Avançon de Nant and Albula sites, while it is non-negligible at the Naviscence. Finally (d), energy propagation across plates was observable for the entire range of A_max_ (≈0.01 and ≈10 V) at the Avançon de Nant site, while it was only detectable for the upper range A_max_ at the Albula and Naviscence sites. A range of reasons can be invoked to explain the variability in plate response across the investigated sites.

A first set of reasons for such differences are related to the design of the experiment itself, and to its operation in the field. While the initial laboratory test showed that the standard deviation of the recorded A_max_ stabilized at a relatively low level when repeating a given impact ≈10 times (Figure 2, Table 2), the different flow conditions surrounding the apparatus during the field experiment may have influenced the energy propagation from the impacting weight to the plate through the hammer base, and thus the *E_k_* estimated from the laboratory experiment. Additionally, the slight support on the hammer stake that was necessary in some instances where flow conditions were too strong, may have affected the signal recording. These aspects notably influence point (a)—the relationship between *E_k_* and the direct A_max_ recorded—and may be related to the greater variability between individual plates that has been observed at the Albula and Naviscence sites (Figure 5, Table 5, Appendix A). When considering points (b), (c), and (d), the surrounding flow conditions are less likely to generate variability in the propagation of energy across plates, because the energy propagation is presumably less affected by the detailed impact conditions at the impact location then by the elastic properties of the SPG structure. 

A second set of reasons are related to the SPG system, and how it is mounted at a given site. First, there is an intrinsic variability in the sensitivity of the individual geophone sensors mounted within the SPG system, with an open-circuit sensitivity of 27.6 V/m/s that can vary by about 10% (Geospace technologies, personal communication). This may account for a part of the variability in signal response observed in our experiments. Second, it has been shown that different torque moments applied to the screws fixing a steel plate to the steel frame may also affect the response of each individual plate. For example, when varying the torque moment applied to the screws from 7 to 30 Nm, the signal response for an impact in the center of the plate remained almost constant in terms of A_max_ but the impulse count increased from 7 to 10, or by about 43 % [51]. Third, the way the steel frame has been embedded during civil engineering works may also influence the response of plates between different segments, and/or at different sites, to an unknown extent [37,38,45]. Finally, some disturbance of the measuring system after construction, notably due to flood events, may also change the sensitivity of individual plates, to an unknown extent. The elements cited above can influence all (a), (b), (c), and (d), and, at this stage, it is difficult to give a hierarchical importance between them to explain the observed variability between plates. 

### 4.3. Consequences for SPG Calibration

Even though the SPG system is exposed to multiple sources of variability, results of the impact experiment have shown that the signal response variability of individual plates remains relatively low within and in between sites. This is notably observable at the Avançon de Nant and Albula measuring sites, where individual SPG responses to energy propagation across plates were shown to be comparable within a given site, and also in between the two sites (Figure 12, Table 7, Appendix A). For these two stations, the transfer of calibration coefficient in between SPGs at a given site, or even in between sites, may be considered. At the Naviscence however, clearly less energy propagated across plates as compared to the Avançon de Nant and Albula measuring sites, and may therefore require the use of different calibration coefficients than at the other two sites.

The transferability of calibration coefficients between plates within and in between sites, as well as the systematic correction of the propagated energy across plates, is challenged by different elements. First, the signal recorded by a given plate for a given bedload particle can differ depending on the particle’s trajectory, which notably depends on surrounding flow conditions, as for example water discharge, flow velocity, and bed roughness [39,40,41,52,53]. For example, Kuhnle et al. [51] experimentally tested an acoustic sensor very similar to the SPG system, using a steel plate with similar dimensions but an accelerometer sensor instead of a geophone sensor [43]. They found that the mass of a grain impacting the plate is related with a power law to the maximum amplitude A_max_ of the signal. The scatter of A_max_ of the experimental data for a given particle mass was up to about a factor of five, and thus in a similar range as for a comparable experimental data set between A_max_ and particle size for the SPG system [39]. Second, there are temporal changes in bedload transport rates across the section, which may change the proportion of signal that propagates across plates at a given time [37,38]. This also applies to calibration samples collected during natural bedload events, because these are usually gathered downstream of a single plate, which could also record part of the signal that is generated by impacts on neighboring plates [37,38,48,49]. Third, it was shown that the location of the impact on a given plate affects the signal response for a given impact [54]. This point is also true for example for the Japanese pipe microphone system [42,55], an alternative surrogate bedload transport measuring technique. Fourth, the shape of a particle also influences the signal response for given flow conditions [56]. For all these reasons, the transferability of calibration coefficients from calibrated plates to non-calibrated ones should be performed with care.

### 4.4. Future Research along this Topic

When considering natural bedload events, it is unlikely that free-falling cylindrical weights used in this experiment are closely mimicking the way particles impact the plates during natural bedload transporting flows. Additionally, the shape of the weights used in this experiment may influence the signal pattern recorded by the plates for each specific impact. Therefore, there may likely be some differences in the signal recorded by the plates between natural bedload events and the patterns determined from the impact experiment presented in this paper. This aspect could for instance affect the relationship between impact A_max_ and the impulse count (>0.1 V) N_imp_ that was established from our experiments (Appendix A), due to different trajectories and particle shape impacting the plates. Further developments in similar experimental systems could therefore consist of impact experiments where weights of spherical shape impact the plates underwater, ideally in a setting that is not affected by the surrounding flow but still in conditions that are closer to natural bedload transport events than the impact experiments of this study. Assessing individual sensor response with varying hydraulic conditions (e.g., discharge, flow velocity, turbulence rate, suspended sediment concentration) would also be insightful.

Since the SPG system records bedload particle impacts at high temporal resolution (10 kHz) and the propagation of energy across plates appears to be comparable to some extent between individual SPGs, it may also be possible to distinguish the signal from direct impacts and the propagated signal, and to systematically correct for it. This could result in a better understanding of the relationship between the signal recorded by the plate and the actual mass of bedload in transport, and may reduce the need for collection of large calibration samples. Finally, recent research aimed to identify bedload particle size classes using acoustic sensors, based on the maximum amplitude of the signal [40,43]. By developing an impact experiment that covers the range of maximum amplitude produced by mixed-size particles during natural bedload transport events, the results presented in this paper contribute to the effort made towards a grain-size identification of bedload transport using the SPG system. Further studies are nevertheless needed in this domain to improve the conversion of the SPG signal in grain-size fractions of bedload, which would help to reduce the field calibration effort that is needed.

## 5. Conclusions

The indirect measurements of bedload transport using acoustic or seismic sensors offer new perspectives for the continuous monitoring of coarse sediment transport, notably in steep stream environments, where such datasets are typically difficult to acquire. 

Impact sensors such as with the Swiss plate geophone (SPG) system rely on calibration and validation data to convert a relative signal of bedload transport intensity into an actual mass flux of coarse sediment in transport. An impact experiment was conducted in this study using artificial weights to test the variability in signal response of 43 individual SPGs located at four field measuring sites in the Swiss Alps. In a second step, the propagation of energy (measured in terms of the recorded maximum signal amplitude) in space from impacted plates towards non-impacted neighboring plates at each site was also quantified. The impact experiment was first tested and developed in the laboratory, and was then applied to the 43 plates in the field. 

The experiments showed that the multiple impacted plates have a comparable signal response in general. The maximum amplitude A_max_ recorded by the SPGs was related to the impact energy *E_k_* by a power law function, with an exponent slightly larger than 1. The propagation of energy across plates could also be expressed by a power law relationship between A_max_ and the propagated A_max_, with an exponent slightly lower than 1. The proportion of energy that propagates across plates appeared to be non-negligible, notably when considering the impulse counts N_imp_ greater than 0.1 V. Energy propagation across plates is therefore an aspect that must be taken into account for a better understanding of the SPG system. Differences in the energy propagation across plates were identified between the Naviscence and the other two sites (rate of attenuation, SPG sensitivity), which means that different calibration coefficients may be required for different sites. There are several aspects related to the SPG system and field installation properties that may be responsible for the observed variability in the signal response of individual SPGs, which need to be further investigated. 

## Figures and Tables

**Figure 1 sensors-20-04089-f001:**
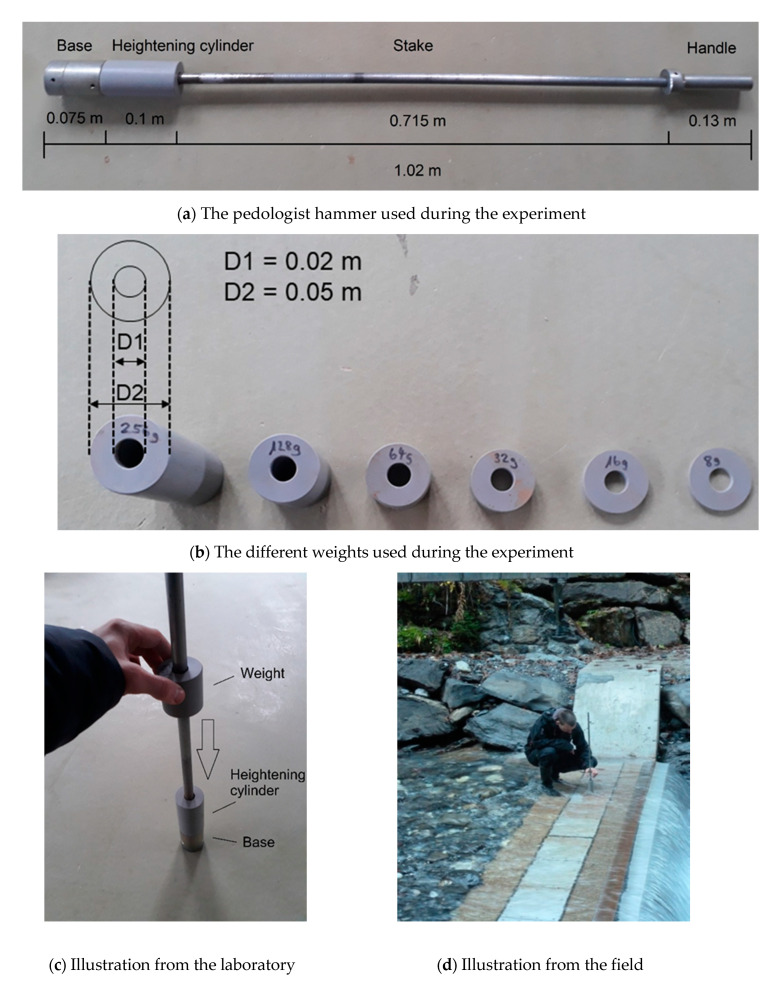
Set up of (**a**) the pedologist hammer, (**b**) the weights used for the experiment, (**c**) illustration of the experiment in the laboratory and (**d**) in the field.

**Figure 2 sensors-20-04089-f002:**
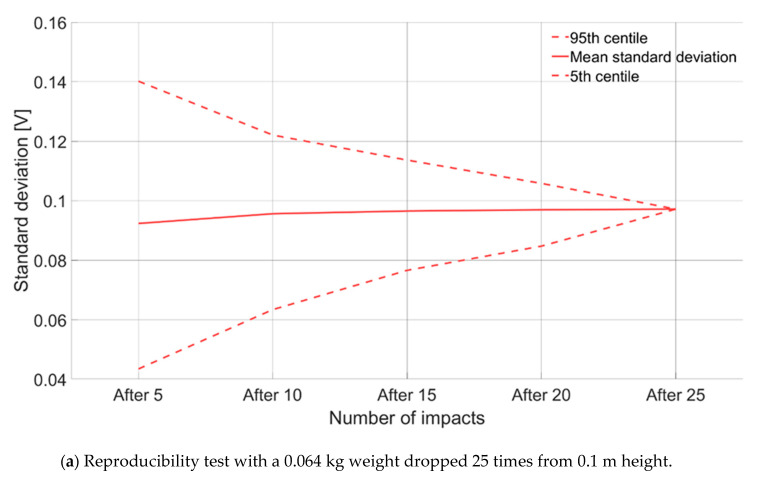
(**a**) Reproducibility test conducted in the VAW laboratory with a weight of 0.064 kg dropped 25 times from a 0.1 m height, the graph shows the standard deviation of maximum amplitudes as function of cumulative number of tests calculated in 5 million random combinations; (**b**) voltage associated to 10 impacts induced by a 0.064 kg weight dropped from a 0.1 m height. The maximum positive amplitude (V) is recorded for each impact.

**Figure 3 sensors-20-04089-f003:**
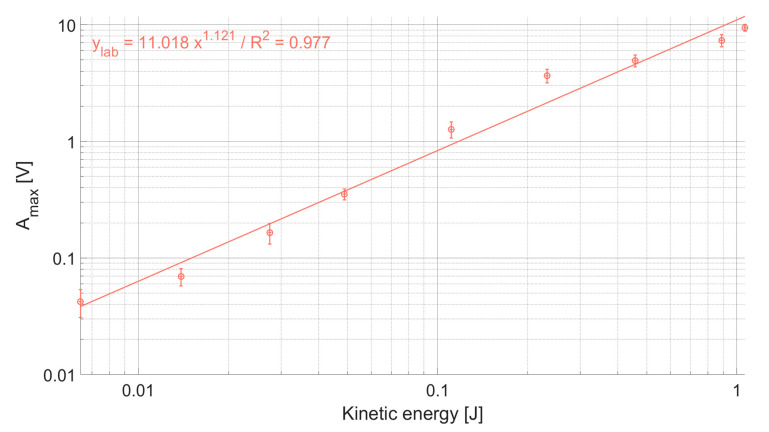
The maximum amplitude A_max_ recorded by the laboratory plate is best expressed by a power law function of the kinetic energy released by the impact, with an exponent slightly above 1.

**Figure 4 sensors-20-04089-f004:**
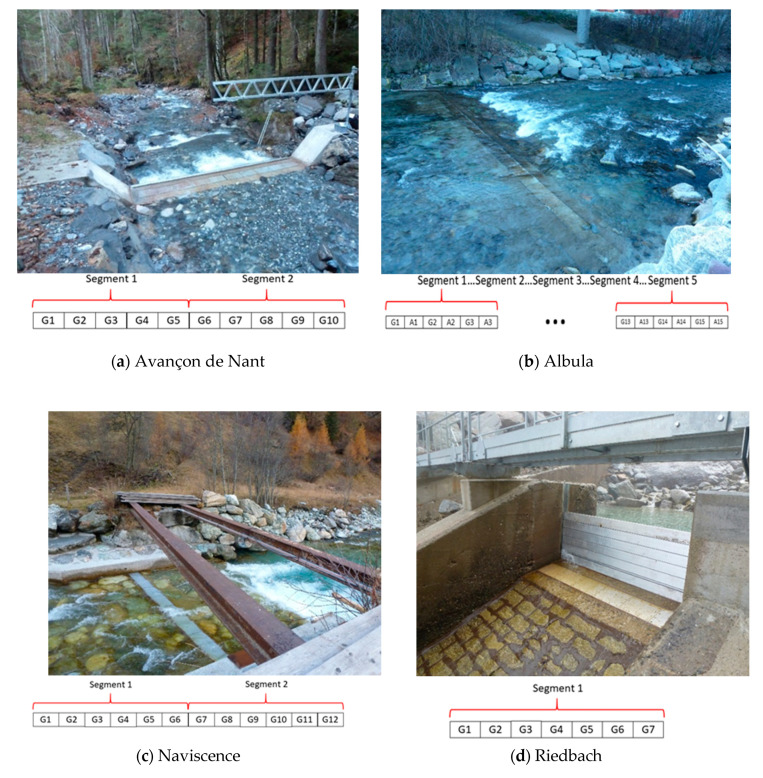
The four measuring stations where the impact experiment was conducted and the setup of the plates and segments: (**a**) at the Avançon de Nant (Western Swiss Alps), (**b**) at the Albula (Eastern Swiss Alps), (**c**) at the Naviscence (Southern Swiss Alps), and (**d**) at the Riedbach (Southern Swiss Alps). The numbers on the plates give the Swiss plate geophone (SPG) number. At the Albula, plates ‘A’ are accelerometer sensors, the others labelled G are SPGs. The numbering is made with view in the downstream flow direction at the Avançon de Nant and Riedbach measuring stations, and the upstream flow direction at the Albula and Naviscence measuring stations.

**Figure 5 sensors-20-04089-f005:**
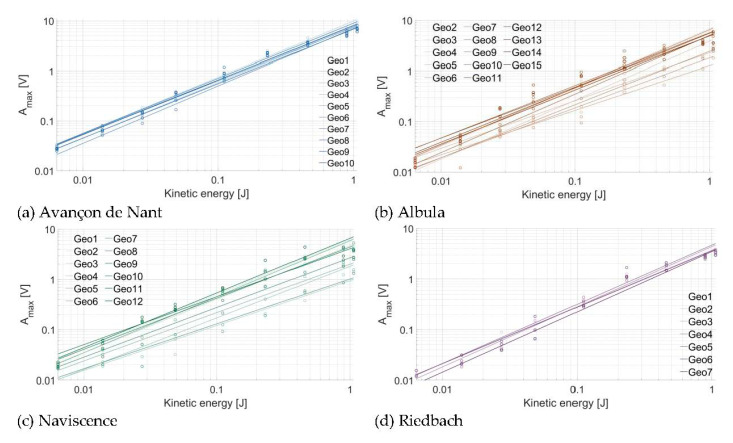
Relationship between the kinetic energy (J) of the impact and the associated A_max_ at (**a**) the Avançon de Nant, (**b**) the Albula, (**c**) the Naviscence, and (**d**) the Riedbach. Each point represents the $\bar{A_{\max}}$ (V) of the given drop category repeated 10 times.

**Figure 6 sensors-20-04089-f006:**
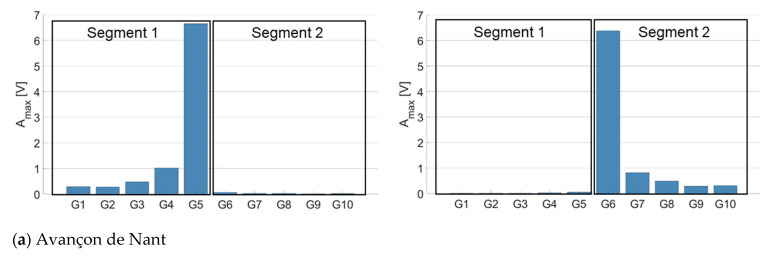
Energy propagation from impacted plates towards non-impacted plates, and across segment limits, at (**a**) the Avançon de Nant, (**b**) the Albula, and (**c**) the Naviscence. Plates located at the margin of different segments have been impacted in this example to illustrate how the energy propagates at the different measuring sites. In this example, one plate per scheme was impacted 10 times (average value) with a weight of 0.256 kg dropped from 0.6 m height (category 9).

**Figure 7 sensors-20-04089-f007:**
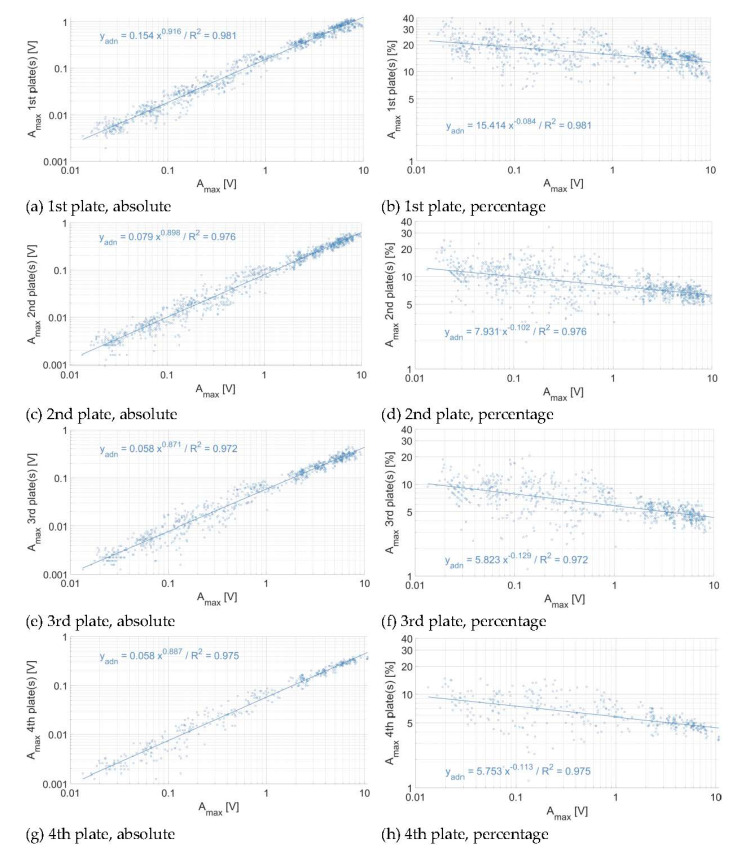
Energy propagation across plates at the Avançon de Nant measuring station (**a**,**b**) at one plate of distance, (**c**,**d**) at two plates of distance, (**e**,**f**) at three plates of distance, and (**g**,**h**) at four plates of distance, in absolute (left column) and relative (right column) numbers. Each dot represents the voltage associated to an individual impact.

**Figure 8 sensors-20-04089-f008:**
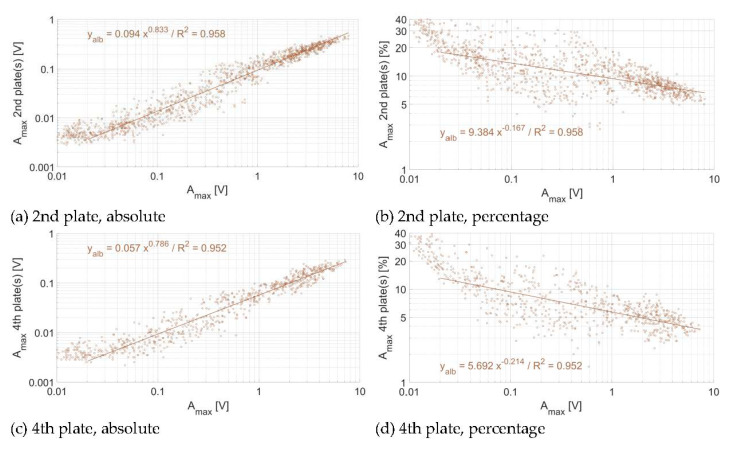
Energy propagation across plates at the Albula measuring station (**a**,**b**) at two plates of distance and (**c**,**d**) at four plates of distance, in absolute (left column) and relative (right column) numbers. Each dot represents the voltage associated to a single impact. The regression is fitted only for A_max_ greater than 0.02 V.

**Figure 9 sensors-20-04089-f009:**
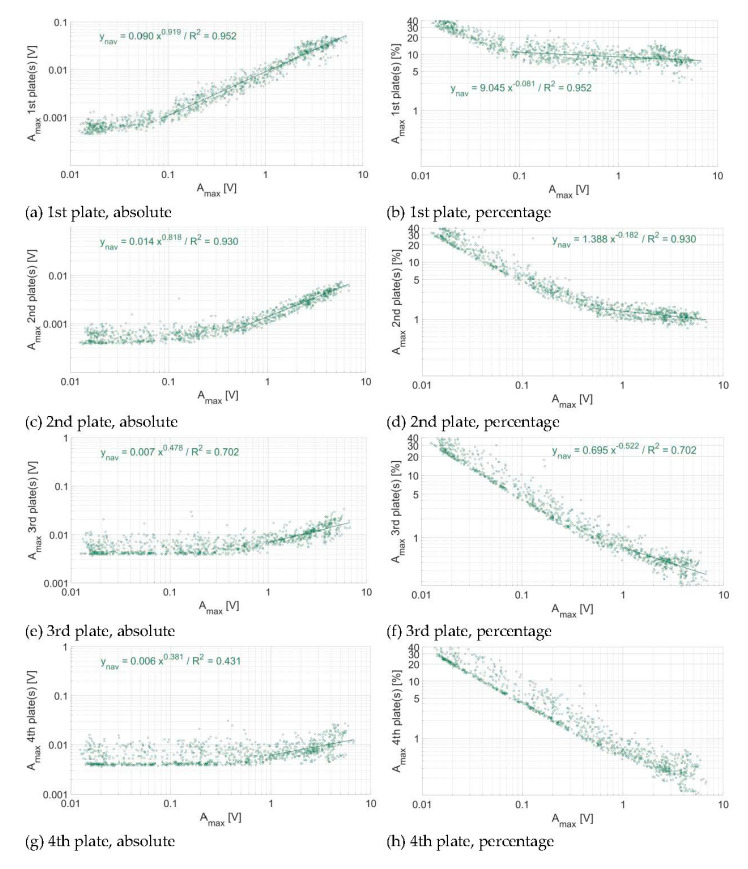
Energy propagation across plates at the Naviscence measuring station (**a**,**b**) at one plate of distance, (**c**,**d**) at two plates of distance, (**e**,**f**) at three plates of distance, and (**g**,**h**) at four plates of distance, in absolute (left column) and relative (right column) numbers. Each dot represents the voltage associated to a single impact. The regression is fitted only for maximum impact amplitudes greater than 0.08, 0.5, 1 V at one, two, three, and four plates of distance, respectively.

**Figure 10 sensors-20-04089-f010:**
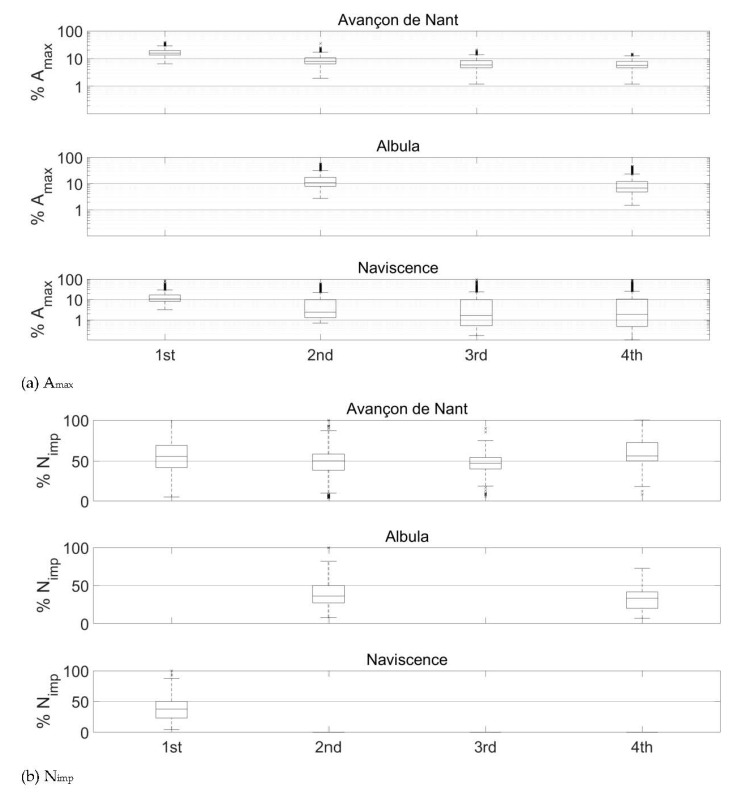
Percentage of (**a**) the impact A_max_ (note the logarithmic scale on the y-axis) and (**b**) N_imp_ (>0.1 V) recorded at one, two, three, and four plates of distance in all three investigated measuring stations. Boxplots show the 5th, 25th, 50th, 75th, and 95th percentiles of the distribution, outliers are marked with crosses. Note that A_max_ covers a range of impacts between 0.01 and 10 V, while N_imp_ (-) only considers the impacts greater than 0.1 V. At the Albula, there is no data available at one and three plates of distance, because there are accelerometers mounted below each second plate. At the Navisence, no impulses are detected further than one plate of distance.

**Figure 11 sensors-20-04089-f011:**
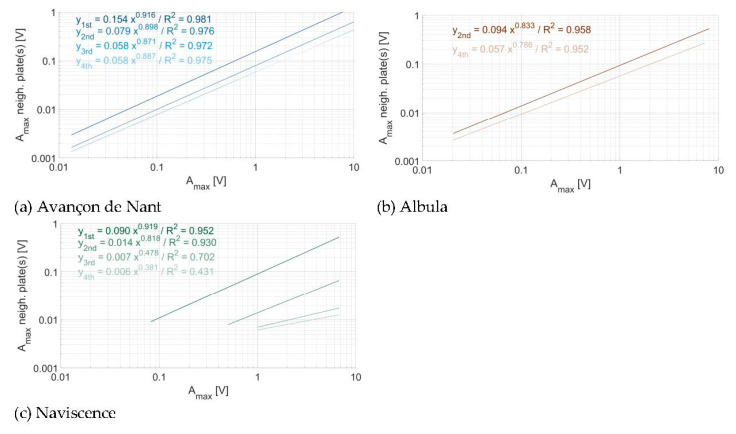
Attenuation of the propagated energy across plates at (**a**) the Avançon de Nant, (**b**) the Albula, and (**c**) the Naviscence measuring stations. Regressions are calculated from all impacts on every plate taken together at each given site.

**Figure 12 sensors-20-04089-f012:**
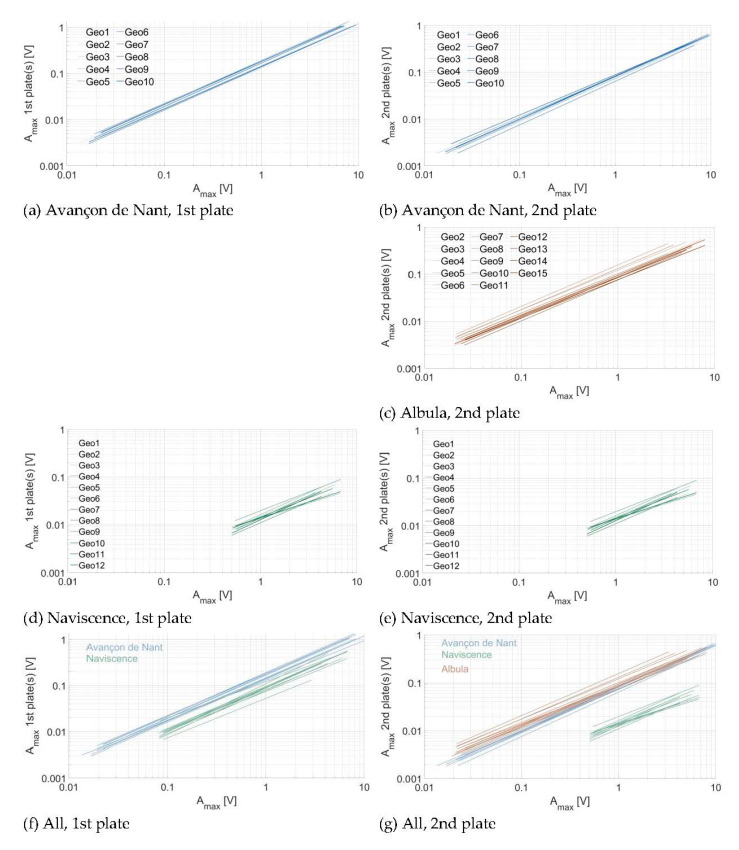
Energy propagation across plates computed for individual plates (**a**,**b**) at the Avançon de Nant, (**b**) at the Albula, (**c**,**d**) at the Naviscence, and (**f**,**g**) for all three measuring stations together.

**Table 1 sensors-20-04089-t001:** Characteristics of the devices used for the experiment.

Element	Mass (kg)	Length (m)	Diameter (m)	Remarks
Hammer (without handle)	2.386	0.89	0.05/0.015	Base/stake
Hammer (with handle)	2.948	1.02	0.05/0.015	Base/stake
Heightening cylinder	0.250	0.1	0.05/0.02	Cylinder/hole
Hammer set up for experiments (without handle + heightening cylinder)	2.272	0.89	0.05/0.015/0.05	Base/stake/height cylinder
Weights	0.256	0.104	0.05/0.02	Cylinder/hole
	0.128	0.051	0.05/0.02	Cylinder/hole
	0.064	0.021	0.05/0.02	Cylinder/hole
	0.032	0.013	0.05/0.02	Cylinder/hole
	0.016	0.007	0.05/0.02	Cylinder/hole
	0.008	0.003	0.05/0.02	Cylinder/hole

**Table 2 sensors-20-04089-t002:** Mean and standard deviation (10 impacts) for each of the nine categories of drop weight-height that were retained from the laboratory experiment to be conducted in the field.

Category	Mass (kg)	Height (m)	$\bar{A_{\max}}\;(V)$	$\sigma_{A\max}\;(V)$	$\sigma_{A\max}/\bar{A_{\max}}\;(\%)$
1	0.008	0.1	0.042	0.011	26
2	0.016	0.1	0.069	0.012	17
3	0.032	0.1	0.165	0.034	21
4	0.064	0.1	0.353	0.038	11
5	0.128	0.1	1.267	0.201	16
6	0.256	0.1	3.661	0.491	13
7	0.256	0.2	4.93	0.581	12
8	0.256	0.4	7.341	0.877	12
9	0.256	0.6	9.442	0.651	7

**Table 3 sensors-20-04089-t003:** Average fall velocity and kinetic energy for each category of weight-height calculated based on high-speed camera video recordings.

Category	Mass (kg)	Height (m)	$\bar{v}(m/s)$	$\bar{E_{k}}\;(J)$
1	0.008	0.1	1.27	0.006
2	0.016	0.1	1.32	0.014
3	0.032	0.1	1.311	0.027
4	0.064	0.1	1.234	0.049
5	0.128	0.1	1.317	0.111
6	0.256	0.1	1.347	0.232
7	0.256	0.2	1.89	0.457
8	0.256	0.4	2.636	0.891
9	0.256	0.6	2.883	1.064

**Table 4 sensors-20-04089-t004:** Number of tested plates at four measuring stations where the impact experiment was conducted. The four measuring stations are all located within the Swiss Alps, at Alpine rivers that give their name to the measuring site.

Measuring Station	Location	Coordinates (WGS 84)	Number of Plates	Total Number of Impacts
Avançon de Nant	Western Swiss Alps	46°15′12′’ N/7°06′35′’ E	10	90
Ablula	Eastern Swiss Alps	46°39′41′’ N/9°34′48′’ E	14	1260
Naviscence	Southern Swiss Alps	46°07′52′’ N/7°37′38′’ E	12	1080
Riedbach	Southern Swiss Alps	46°15′44′’ N/7°55′17′’ E	07	630
Total	Total		43	3060

**Table 5 sensors-20-04089-t005:** Regression coefficients of the relationship between the impact kinetic energy derived from laboratory experiments, and the recorded direct A_max_, for all the impact tests over all the plates (taken together) at the Avançon de Nant, Albula, Naviscence, and Riedbach measuring stations, respectively, and the results for the corresponding laboratory experiment.

	Intercept	Slope	R^2^
Avançon de Nant	8.183	1.107	0.991
Albula	4.528	1.077	0.988
Naviscence	3.867	1.030	0.99
Riedbach	3.963	1.144	0.984
Laboratory	11.018	1.121	0.977

**Table 6 sensors-20-04089-t006:** A_max_ required for the 0.1 V threshold to be exceeded at one, two, three, and four plates of distance in all three investigated measuring stations.

	1st	2nd	3rd	4th
Avançon de Nant	0.6 V	1 V	2 V	2 V
Albula		1 V		2 V
Naviscence	1 V	-	-	-

**Table 7 sensors-20-04089-t007:** Regression coefficients between the impact A_max_ and the associated energy propagation at one and two plates of distance recorded for all the impacts on every plate taken together at a given measuring site.

	1st Plate			2nd Plate		
	Intercept	Slope	R^2^	Intercept	Slope	R^2^
Avançon de Nant	0.153	0.919	0.978	0.079	0.899	0.976
Albula				0.094	0.832	0.958
Naviscence	0.091	0.919	0.932	0.014	0.813	0.903

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
