# Peer review of "Bedload Transport Monitoring in Alpine Rivers: Variability in Swiss Plate Geophone Response"

_sensors, 2020, doi:10.3390/s20154089_

Round 1

Reviewer 1 Report

This topic of this manuscript fits perfectly with a publication in Sensors. The experiments and field campaigns were well designed and conducted. The results were well presented. Accepting with the current form is recommended.

Line 117, Table 1. In the column 'Mass[k]', what is [k]? It might be a typo of kg.
Line 145, figure 2. The authors mentioned a 0.064 kg weight was used in the experiment, but there is no 0.064kg weight in table 1.
Line 177, Table 2, in the column 'Mass[kg]', most values are different from Table 1 (e.g. 0.008 kg in table 2, but 0.08 in table 1). I assume they should be identical to each other. Column title 'Sx Amax[V]' is confusing, because it looks like this column is about the product of Sx * Amax. So it is the column title 'Sx Amax/ Amax'
Line 191 Typo mass unit [k]?
Line 196, Table 3 has the same problem as table 2 when compared with table 1.
Line 200. Figures 3. The Y axis is labeled with Amax[V], but it is more like the mean of Amax instead of Amax. Same problem with some of the following figures.

Author Response

Please see the attchment.

Reviewer 2 Report

Lines 45-62: This section is difficult to read, authors need to re-word sentences and proofread sentence structure for ease of readability.

Lines 66-67: "Over the past 20 years, more than 20 measuring 66 stations in the Swiss and Austrian Alps have been equipped with such a measuring system" -- why do the authors limit the statement to the Swiss and Austrian Alps when these systems have also been used in many other regions in the world? I would suggest including those other regions as well.

Lines 64-77: Authors use the word 'robust' several times - try to report in the introduction and results using less exaggerated wording.

Lines 79-91: The authors need to specify to the reader that they are assessing the variability in signal response and the representativeness of one calibration coefficient across space/distance. It appears that a lengthy time record was not evaluated, and so there may be variability over time between low, moderate, and high magnitude flow events but this study does not examine that specifically.

Lines 95-96: Again, the authors need to be more specific that the aim of the experiment is to study the response variability of SPGs across space, but not focusing on variability over time. For example, if the authors were to study different magnitude flow events specifically, then the results would be more broadly conclusive.

Lines 108-115: need re-wording, not easy to read (i.e. Line 113 "it enables to compare") words are missing here.

Lines 123: This is the first time within the Methods section that the authors mention that an initial laboratory experiment was performed. I suggest revising the methods section, describing the methods in order of completion. Also be more specific specific throughout the paper whether impact tests are being conducted using laboratory weights versus river gravel. This needs to be specified in the beginning of the paper as well.

Line 125: How do video recordings of falling weights give any information about the effects of friction? This is not clear in the paper.

Line 212: "90 impact tests per plate" --> 90 impact tests were conducted per plate

Table 4: How far apart are these measuring stations? Should be described.

Line 320: "We also quantified the extent to which some energy propagates from impacted plates towards 320 non-impacted plates." -- is this because of possible edge-effects due to the fact that the plates are touching one another?

The results section is reasonable and well explained.

Author Response

Please see the attchment.

Reviewer 3 Report

The submitted manuscript presents and discusses sensor performances to capturing bedloads in Alpine rivers. The work is interesting and deserves publication after some minor-moderate revision, in my opinion. Some points can be strenghten which are:

1) an overview of sediment mechanics should be added somewhere in the first half of the manuscript, so that the readers can more directly understand how the sensor performances (from the central part of the manuscript) actually affect the bedload dynamics and its quantification through those sensors;

2) some more applied examples could be added, in the introduction, to further widen the fields of application of, for example, pressure sensors recording large-scale experimental pyroclastic flows (Doronzo D.M., Dellino P. 2011. Interaction between pyroclastic density currents and buildings: Numerical simulation and first experiments. Earth and Planetary Science Letters, 310, 286-292), and natural channeled pyroclastic and debris flows  among others;

3) Lastly, please further clarify how the present results are sensitive, especially in terms of their applications, to grain size distributions of natural sediments, i.e. what about the suspended and washload into the flow?

Best regards
